# Challenging Future Generations: A Qualitative Study of Students’ Attitudes toward the Transition to Animal-Free Innovations in Education and Research

**DOI:** 10.3390/ani13030394

**Published:** 2023-01-24

**Authors:** Lara Andreoli, Ronald Vlasblom, Rinske Drost, Franck L. B. Meijboom, Daniela Salvatori

**Affiliations:** 1Anatomy and Physiology, Department Clinical Sciences, Faculty of Veterinary Medicine, Utrecht University, 3584 CL Utrecht, The Netherlands; 2Institute of Life Sciences and Chemistry, HU University of Applied Sciences, 3584 CS Utrecht, The Netherlands; 3Sustainable Animal Stewardship, Department Population Health Sciences, Faculty of Veterinary Medicine, Utrecht University, 3584 CM Utrecht, The Netherlands

**Keywords:** animal-free innovations, animal ethics, animal testing, challenge-based learning, non-animal methods, translational research

## Abstract

**Simple Summary:**

In February 2022, the university course Replacing Animal Testing (RAT) Challenge was organized for the first time by an alliance of Dutch universities with the aim of engaging future generations in finding solutions to replace the use of animals in testing. Educating the next generations of researchers and professionals is an important step in the transition to promote animal-free science, to which the Dutch government is committed. Students are key stakeholders in the transition to animal-free innovations but their perspective still remains underrepresented. Therefore, our study aimed to explore their viewpoints focusing on their beliefs, values, and the motivations of students to join the course. We conclude that students share the ethical and scientific values that inspire the transition, and that their reflections on the socio-political landscape provide valuable insights on current and future challenges to the acceptance of animal-free innovations.

**Abstract:**

In 2016, the Dutch government declared its commitment to phasing out animal experiments by 2025. Although a high number of animal experiments are still performed and the 2025 target will not be met, the commitment remains. Efforts are being made to identify levers that might foster the transition to animal-free science. Education has been found to play a key role in the future of animal-free science and young generations are increasingly seen as key stakeholders. However, their attitudes toward the transition to animal-free innovations have not been investigated. The present article focuses on the values and beliefs held by students, who in 2022, participated in the course ‘Replacing Animal Testing’ (RAT) Challenge, organized by a consortium of Dutch universities. Contextually, students’ motivations to follow the course were investigated. The research was based on a qualitative study, including semi-structured interviews and a literature review. Our analysis of the findings revealed that students feel aligned with the social, ethical, and scientific reasons that support the transition to animal-free innovations. Moreover, the participants identified a series of regulatory, educational, cultural, and political obstacles to the transition that align with those identified in recent literature. From the discussion of these findings, we extrapolated six fundamental challenges that need to be addressed to foster the transition to animal-free science in an acceptable and responsible way.

## 1. Introduction

Animal experimentation has long been at the center of current public and scientific debates [1,2]. Over time, the reflection on its moral permissibility has given rise to a multiplicity of philosophical stances that today inform the literature on the topic [3,4]. Those stances that called for a limitation of the instrumental use of non-human animals (referred to herein as animals) in research were positively received by European institutions during the last decade. The 3Rs principles of good experimental practice (replacement, reduction, and refinement) [5] first formulated in 1959 have been embedded in the EU Directive 2010/63/EU with a focus on replacement [6] and guide ethical committees in their assessments of animal experiments. The same Directive acknowledges that animals have an intrinsic value and clearly states that no animal experiment should be performed for scientific or educational purposes if these can be reached with animal-free methods (EU Directive 2010/63/EU/art. 12). In 2016, the Dutch government took a leading position in Europe by declaring its commitment to phase out animal testing by 2025. Shortly thereafter, the TPI (Transitie Proefdiervrije Innovatie/Transition to animal-free innovation) Nederlands was created with the goal of boosting and increasing confidence in animal-free innovations within and outside the laboratories [7]. Despite the promise of, and scientific innovation in, non-animal approaches, a high number of animal experiments are still performed, and the figures reported by the Annual Reports (Zo doende) in the last five years have not decreased significantly [8,9,10,11,12,13]. Although the Dutch government’s initial ambitions have not changed, these data suggest that the target objective will not be met in the near future. The Dutch situation reflects wider European trends. In 2019, the total number of animals used in research and testing, routine production, and education purposes covering the 29 countries (EU-28 and Norway) reached 10.4 million [14].

Recent research [15,16] has mapped current barriers to the implementation of animal-free methods, delivering a complex landscape of governmental, regulatory, and societal roadblocks that need to be addressed to facilitate the transition. Inadequate funding for non-animal research, lack of data sharing among research institutions, and outdated regulatory requirements are often identified as important constraints. While technological advancements and policy-driven initiatives continue to occupy a top priority on the transition agenda, scholars conclude that technical solutions alone are ultimately insufficient to support broad systematic changes [15,17]. Lohse (2022), for instance, argues that the ‘inertia’ of the scientific community in proactively supporting the development and sharing of knowledge regarding animal-free methods is partially determined by important socio-epistemic factors such as dogmatic research culture, an overreliance on tradition, and unwillingness to critically revise established practices [17] (p. 44).

Educational progress toward animal-free approaches has often been indicated as a pivotal effort to accelerate the paradigm shift toward human-relevant science. The 2021 EURL ECVAM status report states that “educating the next generation of scientists and regulators in NAM [non-animal methods] and on essential ethical values is an important step toward a fundamental change in society” [18] (p. 58). To pursue this goal, it is also recommended that educational resources and training on the topic are developed not only for graduates, but also for younger students across different educational levels [18]. Fostering the inclusion of animal-free innovations in biomedical and veterinary studies while maintaining a high quality of education is also one of the core objectives of the latest ‘Ambition statement on innovation in higher education’ [19]. This recommends that, when available, animal-free methods should be made mandatory in both education and research as a legal and ethical matter (p. 16). The idea behind both reports is that educating students in animal-free methods will have durable, tangible effects in the future of animal-free science. Indeed, increasing awareness of non-animal methods by future researchers is not only a remedy to scientific inertia, but also to its socio-epistemic causes, whereby education is designed to promote a research culture where animal experiments are not considered the gold standard anymore.

At the present time, education programs within the life science domain offer structured animal-based courses at the under and postgraduate levels, while such courses are more limited for non-animal methods. Notwithstanding, the pedagogical paradigm for which in vivo studies are vehicles of more meaningful learning has been questioned on several grounds. Shapiro (1992) and Merriam (2012) discussed the possible emotional effects of animal experiments in regular, rather than specific educational contexts and their impact on the moral development of students. They posit that inflicting suffering on animals contributes to generating a particular ‘ethos’ where it is right to see animals as commodities, but also to distrust one’s own moral intuitions and circumvent personal convictions [20] (p. 132). More recently, the terms ‘compassion fatigue’ and ‘moral injury’ have been used to describe the unintended consequences of normalizing the use, and in most situations, the killing of laboratory animals. While the first term stands for the risk of becoming progressively insensitive toward human and animal suffering [19], the second describes the physical and psychological burden deriving from moral choices that do not reflect a person’s own moral values [21]. As a result, it has been recommended to protect students’ rights to use animal-free methods wherever possible in the achievement of their learning goals [19].

In a landscape where education in animal-free methods is considered increasingly important, the university course ‘Replacing Animal Testing (RAT) Challenge’ was first organized as a response to this need [22]. The challenge is an innovative educational format that aims to stimulate students from different backgrounds and disciplines to cooperate in finding innovative solutions to the problem of animal testing. Despite the various definitions in the literature, the goal of Challenge-based-learning (CBL) as an educational approach or teaching method is to engage students with current real-life societal challenges in order to “define and address the problem and to learn what it takes to work toward a solution, rather than to solve the problem itself” [23] (p. 1). Working toward the solution is typically conducted in a collaborative, multidisciplinary manner and through the engagement of relevant stakeholders [24]. The course was part of the broader project ‘Challenging future generations’, a partnership among four Dutch academic institutions (Utrecht University, University Medical Center Utrecht, Eindhoven University of Technology, and Wageningen University) in collaboration with the Universities of Applied Sciences [22] aiming at bringing together students, researchers, and professionals to face current societal issues in the health–energy–sustainability field in an interdisciplinary, challenge-based perspective.

This article is the result of a qualitative investigation conducted among the RAT challenge students to understand what young generations (Bachelor and Master’s students between 18 and 25 years old) think about the transition toward animal-free innovations. Following Ezinger and Durnberger (2022), who argue that focusing on young people’s perspectives of scientific practices is particularly valuable as they will be “the future decision makers in society” [25] (p. 2), the aim of our research was to explore students’ attitudes about the transition toward animal-free models. To answer this question, their motivations to participate in the course were investigated. Students’ views on animal testing have previously been the object of scholarly attention both in secondary and undergraduate education [25,26,27]. However, up to now, no research has focused on students’ perception of animal-free methods, a topic that remains underrepresented. This qualitative research, therefore, aims to fill this knowledge gap by shedding light on what views are held by students. We considered the students as key stakeholders in the transition to animal-free science, given their pivotal role as future professionals. We focused on students’ beliefs, motivations, personal values, and values that they would like to see reflected in society, including the scientific community, to identify a way forward in the transition process. Indeed, research has shown that students can be change agents in identifying barriers and gateways to complex social problems [28]. As noticed by Larsen and Parry (2009), in the sustainability field, students have been found to be effective change agents “if they possess knowledge of the environmental, economic and social issues related to sustainability, a value system and motivation” [28]. Therefore, it is important to continue developing courses that educate students on animal-free innovations from a societal perspective and provide useful tools for allowing students to develop critical thinking skills, which is a critical asset in the paradigm shift to animal-free innovations. From the analysis and discussion of the results in the context of previous studies, we could finally extrapolate six fundamental challenges that need to be addressed to facilitate the transition toward animal-free models in a desirable and responsible way.

## 2. Materials and Methods

### 2.1. Research Method

The study took place between May and June 2021 and was conducted via a qualitative study, whereby students were invited to participate in online or in-person semi-structured interviews.

The research method was selected on the basis of the exploratory nature of the research questions, aimed at gaining insights into the values and beliefs of students about the transition toward animal-free models and their motivations to join a challenge-based course on the topic. Individual interviews are particularly suitable in qualitative research for in-depth investigations of people’s perspectives, “personal meanings and experiences of a given phenomenon” as identified by Ryan et al. [29] (p. 313).

A semi-structured format was chosen to give interviewees the space to deviate from the standard questions and possibly explore new related topics [29,30,31]. Moreover, considering the interdisciplinary nature of the course, the semi-structured interview allowed participants to focus on those topics they felt aligned more with their expertise and background.

### 2.2. Study Population

The study was conducted exclusively on those students attending the Replacing Animal Testing (RAT) Challenge.

The Course Guide (academic year 2021–2022) is available in the Appendix A.

### 2.3. Data Collection and Processing

The invitation to participate in the study was forwarded electronically to course participants (n = 37) by the course administrator through the students’ institutional email accounts. Students were able to schedule an online or in-person appointment with the primary researcher (LA). Interviews were performed online using Microsoft Teams, whilst in-person interviews were conducted on campus.

Prior to the interview, students received a consent form and an information letter that outlined the scope of the study, their rights as participants, and detailed information about the data collection, handling, and storage. Students were also informed that their participation in the study would not affect their final grade. Participants did not receive any financial compensation for their involvement in the study.

Interviews were conducted up to saturation point (n = 11) by the primary researcher and were based on a template of 16 open-ended questions, clustered according to four main sub-groups: (1) Background of the participants, (2) Motivations and learning goals, (3) Ethical values and issues of animal experimentation and animal-free methods, (4) The role of education and institutions in the TPI initiative (Table 1). All the interviewees responded to the same questions in the order presented in the template. However, when necessary further questions were asked to clarify a particular answer or to explore a new topic that emerged during the interview.

The interviews were conducted in English and lasted between 20 and 30 min. Each interview was audio recorded and manually transcribed by the primary researcher. All collected data (recordings and interview transcripts) were pseudonymized. The recordings were immediately destroyed after the research team had validated the quality, conciseness, and clarity of the transcripts, which were subsequently stored in accordance with the University guidelines.

### 2.4. Ethics Statement

The research was ethically reviewed by ‘Nederlandse Vereniging voor Medisch Onderwijs (NVMO)’ (NVMO-ERB dossier 2022.3.2).

### 2.5. Data Analysis

The qualitative data (interview recordings and transcripts) were analyzed through a thematic analysis with the scope of identifying recurring patterns or themes related to the core topics expressed in the research questions. The analysis was conducted following the work of Braun and Clarke (2006), who proposed a six-step approach for the identification of themes [31].

In the first phase, the primary researcher read the transcripts repeatedly to become familiar with the dataset. Thereafter, sections or sentences of the transcripts were highlighted and labeled with codes that described core ideas, which the researcher deemed to be relevant to the research. The codes were generated and modified when necessary throughout the coding phase [32,33]. At the end of the coding phase, all related labels were gathered in groups. Those labels that could not be allocated to any group were later set aside for discussion with the research team. The groups were then used to generate themes and sub-themes, which reflected the general patterns emerging from the data analyses. Finally, the data analysis procedure and the related findings were presented and discussed with the research team.

## 3. Results

### 3.1. Students’ Motivations

#### 3.1.1. Learning about Animal-Free Methods

The lack of specific courses on animal-free innovations in the various curricula was identified as being one of the main motivators for students to participate in the RAT Challenge. Indeed, the majority of the students reported not having received any prior education focused on animal-free innovations before the RAT Challenge course. However, all the students with a background in life sciences had received education on animal testing and three reported to have performed in vivo experiments as part of their studies, for example as part of their internships:


*I did an internship in neurotoxicology, so we did an exposure on brain cells that were taken out from a rat pup (Interviewee 4)*



*It was for one course (...) I have done dissections of a few different animals, rats and fish (Interviewee 1)*


Some participants also mentioned having learned about replacement methods, although only theoretically or on occasion as part of an elective or an internship.

Many participants believed that introducing animal-free innovations in formal education is a matter of great importance. They argue that the next generation of researchers will never endorse the applicability and translatability of these models in human-relevant research unless they are familiarized with them early on in their academic careers.


*
Education plays a great role. There is not enough education on [animal-free methods], so that is why people are not inclined to choose them later on in their career. Maybe you learn about them in the internship, but not during your studies (Interviewee 5)
*



*In my curriculum there is no specific course on alternatives. We need a specific course, because if people don’t know they will just use animals. I would have done the same without this course [the RAT challenge] (Interviewee 7)*


The lack of educational focus on animal-free innovation even inspired a group of students within the challenge to develop a specific course on animal-free methods for universities in order to broaden their offer and facilitate being at the forefront of societal change.

#### 3.1.2. Becoming Good Professionals

Contextually, other students reported having felt motivated by the desire of becoming a good researcher or biotechnician, confident that knowing about animal-free models will help them to better substantiate the experimental design of a project and better answer a certain research question.

The interdisciplinary nature of the RAT Challenge, whereby students of different study fields collaborated together, was also mentioned as a learning goal of participants. Students that already came from interdisciplinary studies—such as Science and Innovation management—were eager to prove what they had learned during their academic path:


*My study is focused on combining people, it creates professionals that are bridges between science and society and in this course I can apply the skills acquired during the Bachelor. I want to combine my knowledge with the one of the other team members to use it in society (Interviewee 2)*


However, students who came from a scientific background wanted to learn about other disciplines that also play a role in the transition toward animal-free models, such as policy making and sociology. In this regard, the capability of adjusting one’s own communication style to team members who have different expertise was found to be a recurrent learning goal for students and regarded as fundamental to successful teamwork.


*One of my learning goals is to explain my knowledge to people of different backgrounds, because I always work on projects with engineers. I also want to learn about medical policies. In fact, our project is about policies to improve the implementation of animal-free methods in the field of toxicology (Interviewee 8)*


#### 3.1.3. Values as Motivations

Learning about animal-free innovations functioned as a motivator for most of the interviewed students. For some of them, the desire to learn about the topic is also sustained by deeper reasons relating to their personal values and worldviews. In particular, some interviewees expressed their concerns about the commodification of animals in scientific research and their will to be part of a cultural change whereby animals are not seen as experimental subjects anymore.


*
I’m a vegan myself. I’m against using animals as objects. And I want to work in the medical field. And, in research, there are a lot of animals used. I was just really interested in what is possible to do without them. Is animal experimentation really that necessary as we’re always told? (Interviewee 1)
*



*I found it difficult to accept the idea of having to use animals in research. I am against the use of animals when it is not necessary and I am vegetarian. It was difficult to accept that I would have to use animals in the future. So I hoped to learn but also to contribute to the course with my ideas. (Interviewee 7)*


### 3.2. Animal Testing and the 3Rs

#### 3.2.1. Beyond the 3Rs

Participants reflected on the 3Rs (replacement, reduction, and refinement) principles and their adequacy to regulate animal testing in our society. The 3Rs have played a great role in the ethical progress of animal research in the past and continue to do so since they require researchers to justify why animals are required to be used in their studies. Nevertheless, students identified current pitfalls and limitations of these principles from the perspective of a societal transition toward animal-free testing.

Many believed that the principle of replacement should be given priority over the other two principles. For some students, talking about the ethics of animal experimentation in terms of the 3Rs, implicitly suggests that animal experimentation is still the gold standard, while they believe the scientific community should try to move away from this paradigm.


*
They [the 3Rs] are a great start, but we need to focus more on replacement and alternatives. They are not enough (Interviewee 6)
*



*They are good if you look at them from the perspective of animal tests, especially if considered in the right order: replacement comes first! However, the focus is always on the last two and not really on the replacement. There has to be more emphasis (Interviewee 10)*


#### 3.2.2. The Future of Animal Research

A common belief identified among the students was that animal testing is still necessary for the progress of scientific research, and we, as a society, will probably not be able to conduct research without them in the near future. Animals are relevant to research even when they are not direct experimental subjects, as they are often used as animal-derived components in cell culture media or to harvest cells that will be cultured via in vitro assays.

However, students felt confident that the number of animal experiments will significantly decrease in time and that the critical underpinning of determining the most appropriate research model (animal, as well as non-animal) to answer a specific research question becomes the basis for future research on human health and disease.


*
I thought animals will always be used to study complex systems, but now after the course I am more convinced that at some point they won’t be necessary anymore. Maybe the methods are not that advanced yet and more research needs to be done, but now I feel that the transition will be possible if everyone wants that (Interviewee 7)
*



*Slowly animal experiments will decrease and alternatives will prove to be better models. Think about organs on chips. Animal models are kind of obsolete in a way, they do not translate well to human health (Interviewee 10)*


### 3.3. The Transition toward Animal-Free Models

#### 3.3.1. Ethical and Scientific Values

Participants were asked to reflect on the values that inspire the transition toward animal-free models. Some responded that animal research nowadays is conducted according to ethical standards reflected in attention to high care for laboratory animals and research that is designed to be as pain free as possible to avoid unnecessary discomfort. For students, this was in line with, or a response to, wider public concern for animal welfare in and outside the laboratory.

Students mentioned a variety of ethical principles and concepts that might inform arguments in support of the transition, such as the principle of doing no harm and the principle of benevolence translated in terms of attention to animal welfare. In such cases, students believed that people are increasingly more aware of the intrinsic value of animals’ lives, and thus the ethical significance of shifting toward animal-free research wherever possible.


*We need to sensibilize researchers to look for alternatives not only because they are better scientific models, but because we should change the way to do research. We care about animals more, we know that they are more complex than we think, they are not machines but they have emotions (Interviewee 7)*


Finally, some participants pointed out that there are also scientific reasons in support of animal-free models, since animal experiments do not always translate successfully to solutions applicable to human health. Animal-free innovations can, therefore, help us conduct human-relevant research and find answers to our research questions.

#### 3.3.2. Ethical Concerns

However, a few students were also conscious of the ethical challenges that animal-free innovations have to face. Three main concerns were identified: (1) the procurement and handling of human biospecimens (cells, tissues) used for in vitro models; (2) the creation, in the future, of increasingly complex organoids or multi-organ on-chip systems that could in a way resemble in vivo organisms in some morally relevant features, e.g., sentience or consciousness; and (3) the clinical usefulness of computer predictions which are based on biased datasets (e.g., Caucasians, males).


*With cell cultures it is possible to create more complex organoids. Neuro-tissues might develop sentience and feel pain. They might be not even close to animals now, but if you want to create a system that has a brain it might get close to something conscious. So you might incur the same problem of animal experimentation (Interviewee 11)*


In addition, one of the participants noted that animals are still part of the research chain even if we do not see them, such as in the development of in vitro models where cells are often cultured in animal-derived gels and sera (e.g., Matrigel, fetal calf serum).


*To culture organoids, you need stem cells. This topic could be ethically challenging. OoC, or computer models, make use of a lot of data and they are usually taken from white middle-aged men. What about females and minorities? Research might still not be relevant for society (Interviewee 8)*


### 3.4. Institutions and Public Policies

#### 3.4.1. Structural Constraints

Despite the ethical and scientific desirability of the transition toward animal-free innovations, according to our participants’ study, there are a series of structural constraints that need to be addressed. In their view, scientific research is currently running faster than legislation. For instance, although animal-free models to test new drugs are being developed in laboratories, regulators still require the same drugs to be tested also in vivo.


*Regulations need to catch up. I want to use a replacement method so as not to test the drug on an animal, but I cannot do it according to law. So the tool cannot be used in the public sphere because of legal barriers (Interviewee 5)*


Moreover, the assessment procedure for the approval for use is, according to some participants, a great hurdle to their recognition and effective implementation, especially at a European level.


*
There are a lot of great ideas, but it’s like a forest if you want to get them to use: you don’t know where to go (Interviewee 1)
*



*I read about the policies in different EU countries. There is extensive policy on the topic, however it is so vague. You do not see clear goals, just advice (Interviewee 9)*


A group of students turned this particular issue into their challenge: in order to assist researchers to have their animal-free models approved, they designed a roadmap illustrating which steps need to be undertaken during the various phases of their model’s assessment process.


*We are currently working on a project that tries to make visible what you have to do to actually get your animal-free model approved. And it is a chaos, there needs to be a plan to facilitate that. We are making a roadmap to facilitate this process (Interviewee 6)*


#### 3.4.2. Ensuring Trust

Some students felt that such regulatory requirements would discourage scientists from using animal-free methods, due to complicating their validation and therefore eventually undermining trust.


*
Institutions and governments should do more. If regulations are not updated yet, then animal free methods cannot be used so much. And because they are not used they cannot be properly validated, therefore used, and the circle goes on. The government could break it and invert the trend (Interviewee 8)
*



*Now the gold standard is animal experimentation and scientists do not want to see their research as less valid because they did not use an animal model. Alternatives need to have the same status as animal experiments, so that we can use them without risking that your research will not be published. The more we use them, the more we can prove they are valid (Interviewee 11)*


Students are aware of the critical as well as the delicate role played by the government and the importance of adequate public policies that support the transition toward animal-free innovations from a legal point of view. Some believe that the government should take a stronger position in favor of animal-free models by changing regulatory requirements or even prohibiting animal experiments. On the other hand, other participants warned of the risks of adopting drastic measures and suggested a positive approach, whereby the government acts as a facilitator by ensuring that animal-free models abide by high scientific standards and can be trusted, both within and outside of the scientific community.


*
Policies and rules can influence how science is performed. For example, in drug development: [...] if you say animal testing cannot be used, then people won’t (Interviewee 5)
*



*
Government has a big role but they are not doing enough. Society needs to be involved. Also, the number of animal experiments is not going down. Is it really a priority on the agenda? (Interviewee 2)
*



*Different scientists have different aims. Some want to be ethical and work in the best way, others prioritize the results they can get. If the government starts regulating more heavily and makes procedures that scientists have used for a long time not allowed anymore, you might have scientists moving to other countries to continue research [...] Prohibiting animal experimentation won’t work out. But if you show that animal-free models are valid and your research is valuable, it would help. Many researchers are within a culture and they think that animal experimentation is the only way. The government needs to make sure that validation is regulated, trustworthy, so that more researchers will be motivated to use animal-free innovations (Interviewee 11)*


## 4. Discussion

Our results indicate that students considered it important to learn about animal-free innovations during their educational path. The lack of education offered in the various curricula motivated these students to join the RAT Challenge almost unanimously. It is worth noticing that such motivation was also reported by students who did not have a background in the life sciences. This suggests that there are some students who are aware that the transition regards society as a whole and are willing to contribute to it with different expertise. Whilst education on particular animal-free techniques is currently available in some curricula, our participants and other studies report that it is not mandatory [15]. Furthermore, universities still lack courses that focus on developing critical skills regarding the positioning of animal-free innovations and experimental design. Knowledge on how a model works and how to apply it might be insufficient at the time of choosing what is the best model (animal or animal-free) in a particular context. Model choices in scientific research do not happen in a vacuum. Research practices are human products, and as such, they have their own historical dimension: they are established, accepted, and fixed as paradigms for certain timeframes and animal research is no exception. In fact, many students were well-acquainted with scientific, legal, and ethical aspects of animal experimentation, which they had previously learned in their study programs. In a recent study, Veening- Griffioen et al. (2022) analyzed the arguments put forward by researchers to justify the selection of particular animal models and concluded that in most cases the explanations were tradition-driven, while “it remained unclear whether the selected animal model was the model with the highest likelihood of predicting the clinical outcome” [34] (p. 60).

Performing systematic investigations of all the available animal-free methods in the preparation phase of a study, however, might be complicated by a variety of factors, especially at the early stages of a researcher’s career. Research has proven that time constraints for completing a study and the so-called publication pressure negatively influence the quality of the research and lead to questionable research practices [35,36,37]. Krebs et al. (2022), furthermore, have shown that scientists feel compelled to include animal data in their research only to satisfy reviewers’ requests and increase their chances of being published, a phenomenon they called ‘animal-reliance bias’ [38].

A cultural shift toward better practices in preclinical study design is a matter of integrity and must involve the education of future researchers. Learning how to choose the best models to answer research questions should therefore start early in the curricula and be brought to the attention of the transition agenda for two reasons. First, to increase students’ familiarity with animal-free innovations and their possible applications; and second, to minimize the risk of animal experiments being performed out of bias and habit.

This capability of making correct, informed decisions was also identified by some participants as a quality of the ‘good scientist’. Epistemic choices in research are necessarily interrelated with ethical questions. Ethical reflection in the context of animal experimentation is usually conducted in terms of harm–benefit analysis, which aims to identify what courses of action will maximize societal benefits while minimizing pain and distress for animals. However, less emphasis is placed on those qualities or character traits (virtues) that would make scientists ‘good’ and guide them “in a responsible practice of science” [39] (p. 77). In 2018, the EU Horizon 2020 Project VIRT2UE was launched with the scope of developing a virtue-based approach to research integrity and identifying those moral virtues that would help scientists in meeting the ethical challenges of their job [40]. This suggests not only a possible integration of aspects of virtue theories in ethics modules for life scientists but also that the problem of model choice in experimentation can be tackled by an interdisciplinary approach.

Students’ motivations to participate in the course were not only linked to educational goals but often sustained by values that informed their general worldview and life choices. The concept of value is commonly and intuitively grasped as something that individuals regard as important or worth being pursued. As noticed by Warnock (1996), values are also grounded on the assumption that—unlike preferences—they can be shared in the public arena and accepted by a larger community [41] (p. 46). The relation between motivations and values has been the object of scholarly attention and clarified by the well-known theory of values of Schwartz (2016), who defines values as “broad, motivational constructs that express what is important to people”, informing their life choices and their identities [42] (p. 63). In this framework, one of the features of values is that they motivate human actions and the pursuit of related goals. The testimonies of students in this study demonstrated that the value of limiting animals’ reification across the research and food chain motivated life choices such as being vegan or vegetarian as much as participating in the RAT Challenge and learning about animal-free methods.

Values as motivations, therefore, are particularly significant in the context of societal transitions where—as underlined by Interviewee 9—technological innovation alone is not sufficient to produce long-term structural changes, but people’s choices and attitudes play an important role as well. In this regard, students believe that concerns about the treatment of animals in experimentation, their welfare, and the ‘righteousness’ of their instrumental use are shared by the public and support the ethical desirability of the transition. In fact, such concerns are compatible with a variety of arguments and value frames in pluralist societies: from the utilitarian idea of maximizing societal happiness beyond species distinctions to the belief that animals possess an intrinsic value and their use in research is not ‘right’ per sè—even when the level of discomfort is minimal. As reported by the Nuffield Council on Bioethics (2005), although people might disagree on what particular features make animals worthy of moral consideration (sentience, being subjects of a life), a consensus can be reached at this level: animals possess some morally relevant features and, for this reason, their commodification is highly undesirable [43] (p. 41).

Students also reported that the transition toward animal-free models responds to the scientific exigency of finding better answers to research questions, since animal models fail to deliver informative results to benefit human health [44]. This is in line with a conspicuous part of the scientific literature, which states that the translation of results from preclinical animal-based studies to the clinical phase is frequently unsuccessful, raising questions about the scientific and ethical acceptability of animal experiments [45]. Pippin et al. (2019), for instance, discuss how the scarcity of advancements in animal research of neurodegenerative diseases does not compensate for the material and intellectual resources that have been invested in the last decades [46]. Pound et al. (2018) have attributed translational failures to two main causes: first, animal experiments are too often poorly designed and biased, which has given rise to the so-called ‘reproducibility crisis’ [44,47]; second, results from animal experiments do not always apply to the humans due to important biological asymmetries, as proven by a history of unsuccessful clinical trials [48]. In 2010, the ARRIVE (Animals in Research: Reporting In Vivo Experiments) guidelines were published with the scope of helping researchers to report their animal experiments in a more transparent and methodologically rigorous way to increase reproducibility [49]. Nonetheless, the study of Leung et al. (2018) has shown that although journals support the ARRIVE guidelines, reporting standards of animal experiments have not improved significantly and the description of potential bias factors remains low [50].

The interviews conducted in this study revealed that students are convinced about the reasons that support the transition and confident about its realization; however, they also expressed skepticism about a quick shift toward animal-free science. Despite the ‘epistemic criticism’ of animal experimentation [51], the scientific value of animal experiments remains debated in the literature. Singh and Seed (2021) suggest that animal models can still contribute to our understanding of complex pathophysiological processes in the human body and are essential to drug testing: translational failures and high drug attrition rates are a reality, but they indicate that current animal models—far from being abandoned—should be improved and possibly combined with in vitro and in silico methods [52]. The value of animal experiments is also acknowledged in the modeling of complex inter-system interactions, such as the role of the immune system in the gut–brain axis, which cannot yet be replicated even in the most recent organ-on-chip technology [53].

For as long as animal experiments are performed for scientific reasons, therefore, it is important that they abide by high scientific and ethical standards, so that unnecessary suffering—in quantity (reduction) and severity (refinement)—is avoided. However, the persistence of a high number of animal experiments has also been attributed by students to a matter of principles, namely the insufficiency of the 3Rs to foster the transition toward animal-free models and to centrally position the principle of scientific validity. Participants who called for more attention to the principle of replacement noticed that its axiological and conceptual priority is not reflected enough in practice. However, the 3Rs—and the principle of replacement itself— also exemplify the well-established paradigm where animal experiments are the ‘gold standard’; that is, the point of departure of any study design. Changing this paradigm, on the other hand, would mean including humans already in the initial stages of model design and turning to animals only as a last resort. Ritskes-Hotinga et al. (2020) reported that increasing patients’ participation in preclinical research can help in identifying more relevant research questions and research priorities, which can be addressed without returning to animal studies [54,55].

Besides being promising research models, the students in this study perceive animal-free innovations as less ethically problematic than animal experiments. In fact, very few interviewees were able to mention or reflect in depth on the ethical limitations of animal-free methods. Only one participant pointed out that the use of animal-derived sera to stimulate cell growth in in vitro models does not make them actually ‘animal-free’ as it would appear or is even advertised. Indeed, some of the most common sera in which cells are cultured come from laboratory mice (e.g., Matrigel) or from fetal bovines (FBS or FCS). In the first case, the Matrigel is derived through the induction of sarcoma tumors in mice [56] (p. 1), while FBS is obtained from collecting the blood of at least three-month-old bovine fetuses, which are found accidentally in cows destined for the slaughterhouse [57] (p. 100). The harvesting of fetal blood typically occurs through a cardiac puncture without the use of anesthetics whilst the fetus is still alive. The entire process of blood collection has been accurately described by Jochems et al. (2002), who focused on the limitations of FBS over 20 years ago [58]. The fact that students are not conscious or do not regard animal-derived sera as an ethical issue means that more awareness and more transparent communication on the topic is needed to avoid breaches of public trust toward animal-free innovations, but also to stimulate the development of valid alternatives.

Finally, all of our participants have highlighted the presence of one or a series of structural constraints—here, intended as social, political, or economic obstacles—to the transition to animal-free science. Students understand that societal innovations are complex phenomena that extend beyond academia and research institutions. In the literature on innovation and societal transitions, there is a consensus that innovations need to be accepted in broader ecosystems, which comprise multiple actors and whose cooperation is fundamental to the achievement of long-term societal shifts toward more sustainable practices: different stakeholders, regulators, and funding bodies, as well as politics and culture, play a role [59]. However, since innovations disrupt well-established practices, they regularly encounter “deliberate and passive resistance from different types of actors and institutions” [60] (p. 6357).

In this framework, students reflected on the validation process of animal-free methods and current regulatory requirements. The first was considered too complicated and legally tortuous, a fact that would discourage scientists from undertaking the assessment path and implementing them in their research [61]. Difficulties at the validation stage, furthermore, were thought to have a negative impact on current regulations, which do not consider animal-free methods safe enough to skip in vivo tests for toxicity assessments. Interviewee 8 (cf. 3.4.2) used the image of a vicious circle to describe how the two hurdles feed each other contributing to the obstruction of the innovation ecosystem. The same constraints are also reported as ‘barriers’ in the recent study of Abarkan et al. (2022). In their study, they also found that despite the EU Directive prohibiting animal experiments when suitable alternatives are available, its reinforcement is practically undermined by a lack of clarity about what counts as ‘suitable’ [15] (p. 6).

According to other students, structural obstacles might generate skepticism toward animal-free methods at different levels, among researchers and the public. Steinbruch et al. (2022) have investigated the role of trust in innovation ecosystems and shown that trustworthy relationships ensure more collaborative and proficuous interactions among stakeholders leading, eventually, to the achievement of better results [55]. An exhaustive discussion of trust in the transition toward animal-free innovations goes beyond the scope of this article. However, the topic deserves to be better explored in light of previous studies that describe trust as a multifaceted concept that involves different dimensions of a relationship between parties. Steinbruch et al. (2022) consider the three most discussed dimensions in the literature as ability (technical competence and expertise), benevolence (genuine intention to act in the interest of the other party), and integrity (sharing the same principles and coherence between principles and actions), to conclude that no single dimension alone “is enough to ensure trust in innovation ecosystems, so they must be considered in conjunction” [62] (p. 202).

Finally, students pointed to the delicate role of the government in triggering a virtuous circle of innovation. Some participants called for a stronger, or direct intervention, of the government to switch to animal-free methods. However, Interviewee 11 foresaw the risk that animal research could simply move abroad if Dutch law prohibited it. The study of Bressers et al. (2019) demonstrates that this is a real risk: in their survey, the majority of researchers declared to be ready to emigrate in order to continue their research if stricter legislation against animal experiments were adopted. The same study showed that researchers felt concerned about the position taken by the government in 2016 and suggested that the goal of switching to animal-free science should be coordinated at an international level [16]. In general, however, most of the interviewed students in this study and the literature [15] suggested that ensuring proper validation of animal-free models should be the preferred pathway to have them accepted within the scientific community and at a regulatory level.

## 5. Conclusions

The aim of this study was a first attempt to map the values and beliefs of the RAT Challenge students regarding the transition to animal-free innovations and their motivations for joining the course. Our findings revealed that the students were inspired to participate in the course because they felt aligned with the ethical and scientific reasons that support the transition to animal-free science and believe that education has a key role in promoting such values to foster societal change. Furthermore, students shared their views on current scientific, ethical, legal, and political complexities that characterize the acceptance of animal-free methods in a broader ecosystem. From the discussion of these complexities in the context of current scientific literature, we finally outline six fundamental challenges (Table 2) to a responsible transition to animal-free innovations in research and education.

## Figures and Tables

**Table 1 animals-13-00394-t001:** Questionnaire.

Clusters	Questions
Background of participants	Are you a student and/or a professional? What is your current study program and/or job? What is your educational and/or professional background?
Motivations and learning goals	4. Have you ever performed in vivo experiments for your study/job? 5. Did you learn about animal-free methods during your previous education? 6. Why did you decide to participate in the course RAT Challenge? Explain your motivations 7. What do you expect to have learned by the end of the course? 8. What are your personal learning goals?
Ethical values and issues of animal experimentation and animal-free methods	9. To what extent do you think animal testing will be necessary for the future to progress scientific research? 10. Do you find current ethical guidelines that regulate the practice of animal experimentation (3Rs) satisfactory? 11. Replacing animal testing with alternative methods is an answer to an ethical problem that regards the way we should treat animals. What are, if any, the ethical issues that might arise from the development of new replacing technologies/techniques? 12. Why do you think that the TPI is/is not a relevant initiative to the ethical progress of scientific research? 13. What are, according to you, the values that inspire the TPI initiative?
The role of education and institutions in the TPI initiative	14. According to you, what is the role played by the scientific community and culture in the recognition of animal-free models? 15. Institutions can be at the forefront in boosting scientific innovations. Do you think that more should be done at this level to support the transition toward animal-free science? 16. How can education affect the transition toward animal-free models? Think of your current or previous curriculum and suggest what can be done at this level.

**Table 2 animals-13-00394-t002:** Six challenges to a responsible transition to animal-free innovations.

Themes	Sub-Themes	Challenges
Students’ motivations	Learning about animal-free methods Becoming good professionals Values as motivations	Rethinking higher education on animal-free innovations addressing the problem of model-choice
Animal testing and the 3Rs	Beyond the 3Rs The future of animal research	2. Promoting a paradigm shift in preclinical research starting from study-design
The transition toward animal-free models	Ethical and scientific values Ethical issues	3. Developing clear ethical and legal guidelines for animal-free innovations
Institutions and public policies	Structural constraints Ensuring public trust	4. Guaranteeing and facilitating the validation of animal-free innovations through public policy 5. Updating regulations 6. Building trust among relevant stakeholders

## Data Availability

The data presented in this study are available on request from the corresponding author.

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
