# Peer review of "Challenging Future Generations: A Qualitative Study of Students’ Attitudes toward the Transition to Animal-Free Innovations in Education and Research"

_animals, 2023, doi:10.3390/ani13030394_

Round 1

Reviewer 1 Report

Page 14: Appendix A
The Course Guide of the RAT Challenge (academic year 2021/2022) can be downloaded here: THE LINK IS MISSING!!

References are SLOPPY  and should be reviewd critically!!
#12 no year given!!
For more than 10 references the year is not given in BOLD but regular!! e.g. ##1, 2, 14, 25, 31, 34-37, 39, 44 ,45, 53, 54

Author Response

Dear reviewer,

Reviewer 2 Report

Overall this is an interesting paper worthy of publication. I have a variety of minor comments below. My most significant resquest is for the authors to add material into the introduction explaining what the main barriers to alternatives are, and thus why it is important to educate the next generation of scientists about alternatives - which is important information for understanding why we should care about the views of students. There is considerable material on this subject in the discussion section, but a little is needed upfront to explain the purpose of the paper. For example, the authors in the discussion reflect on how institutional norms are a key barrier to alternatives, since animal models are still typically considered the ‘gold standard’. This is a good argument for why getting the next generation of scientists on board is important - they can thus help to change professional norms in their field. In short, I suggest a paragraph or two in the introduction explaining 1) what the key barriers to alternatives are (e.g., institutional norms), 2) how some of these barriers might be overcome through educating the next generation of scientists, and 3) why we should therefore care about the views of students, and about courses teaching about alternatives.

Minor comments:

-       Simple summary - ‘and their reflections on the socio-political landscape and provide valuable insights on current and future challenges to the acceptance of animal-free innovation within and outside the laboratories.’ This sentence doesn’t make sense currently - I think the second ‘and’ needs to be removed.

P2

-       The first two sentences of the paper are a bit weak. I’d suggest including a reference or two, and rewording to make a bit more specific rather than references to a ‘vast literature’ and different opinions. E.g. Could you talk about protests against animal research, and/or how anti-vivisection groups champion alternatives?

-       ‘nearest future’ - rephrase to ‘near future’.

-       I think the intro needs some background about what the main barriers to alternatives have been - is familiarity with methods or institutional culture for example important? That would help explain why education is thought to be important. This paper might be useful: ‘Scientific inertia in animal-based research in biomedicine’, Simon Lohse, Stud Hist Philos Sci, 2021.

-       Paragraph beginning ‘moreover’ - this could be better introduced to make it clear that you are talking about the use of animals not in research, but in university-level teaching.

-       ‘young students’ -what age group are you talking about here? I’d thought the focus was on university-level students.

-       ‘against one’s own will’ - what does this mean? I think this might need rewording.

-       ‘become in use’ - change to ‘been used’

-       ‘two unintended consequences of’ - this is a bit confusing, I’d remove the word ‘two’ in this sentence.

-       ‘the students’ right’ - remove ‘the’.

-       ‘increasingly more’ - remove ‘more’

-       Is the course name ‘Replacing Animal Testing’ or ‘Replacing Animal Testing (RAT) Challenge’? I would have thought the latter. If that’s not the case, I would remove the word ‘challenge’ after the course name and instead in the next sentence say something like, ‘The course followed an innovative ‘challenge’ format, which aims to stimulate…’

P3

-       ‘Utrecht University and the’ - replace ‘and the’ with a comma

-       ‘in an interdisciplinary’… replace ‘in’ with ‘through’

-       ‘Despite the various… stakeholders’ - I would move this up to after the sentence beginning ‘The challenge is a…’ or vice versa (move the earlier sentence down). Or, the earlier sentence could be cut. Either way, you should discuss what a ‘challenge’ is in just one place.

-       ‘young generations’ and ‘students’ - you need to specify in this paragraph, and ideally earlier as well, what age groups you are working with so the reader knows what you mean by these terms.

-       Add comma after ‘Ezinger and Durnberger (2022)’

-       ‘scholar attention’ - change to ‘scholarly attention’

-       remove ‘in fact’

-       the last two sentences on p3 are each their own paragraph - combine them to make one slightly longer paragraph.

P4

-       It would be useful to say if students were informed that their choice to participate (or not) would not affect their grade for the course.

-       What do you mean by ‘saturation point’ here? Usually that means that you would not find any more significant themes through doing more research, but perhaps you mean that only 11 students volunteered to participate? Please clarify.

-       Table 1 - there’s a skip from 8 to 10 in the list.

-       I notice that in Table 1 there were no questions about what barriers students perceived to the development of alternatives, although this subject did appear to come up based on the discussion. How did this topic of barriers arise in interviews?

P5

-       Paragraphs beginning ‘The qualitative data’ and ‘The analysis’ are very short, just one sentence each. Combine both with the paragraph beginning ‘In the first phase’ to create one larger paragraph.

-       The description of data analysis is overly long - the description of inductive thematic coding could be summarised in just a couple of sentences I think.

-       Ethics statement - I would move this up to the ‘data collection and processing’ section, as that is where consent is discussed.

-       ‘animal-testing’ - remove hyphen

P6

-       I don’t think that these top two quotes add anything to the analysis - it is just students reporting that they have done this previously. More useful would be to give an indication of how many students said they had done some in vivo experiments before rather than ‘a few’, ‘some’ etc.

-       ‘participants believe…They argue’ - I would make these statements past tense for consistency.

-       ‘early-on’ - remove hyphen

-       some students went on to ‘develop a specific course on animal-free methods’ - how did this differ to the course you’re describing in the paper? It’s not obvious how it would differ.

-       ‘To this regard’ - this phrasing is awkward

-       I note inconsistency in the spacing between quotes, and between the authors’ words and quotes - please make sure this is consistent throughout the paper.

P7

-       This connection to veganism/vegetarianism is interesting. This has also come up as a justification for why people volunteer for clinical trials - this connection could be worth noting at some point. See Vanderslott et al., ‘Co-producing Human and Animal Experimental Subjects: Exploring the Views of UK COVID-19 Vaccine Trial Participants on Animal Testing’, STHV, 2021.

-       ‘golden standard’ - ‘gold standard’

P8

-       modify sentence in first paragraph of 3.31 as follows: ‘in line with, or a response to, wider public concern for animal welfare in and outside laboratories’

-       ‘no-harm’ - remove hyphen

P9

-       ‘on a European level’ change ‘on’ to ‘at’

-       The last quote on this page seems to be about scientific inertia and standards within the community - see the suggested reference from Lohse. I also see you mention this in the discussion. I would highlight this a bit more clearly at the time you bring up the quote - ‘one participant highlighted the importance of traditions and norms in science…’ or something.

P10

-       ‘a part of students who is well aware’ - should be something like ‘there are some students who are aware…’

-       ‘cannot overlook the education…’ Do you mean something like ‘must involve the education’?

P11

-       ‘intended by some participants’ - should intended be identified?

-       ‘aims at identifying’ - ‘aims to identify’

-       ‘scholar attention’ should be ‘scholarly attention’

-       “animal’s reification” - should be “animals’ reification”

-       ‘discusses’ after Pippin et al should be ‘discuss’

P12

-       ‘the human species’ - change to ‘humans’

-       ‘last resource’ - last resort?

-       ‘recurring to animal studies’ - recurring is the wrong word I think. Returning to? Or just remove the ‘to’ after recurring?

-       This is an interesting point about how ‘animal free’ doesn’t really mean animal free, and a lack of awareness about the ethical issues around such methods. This resonates with work by Gorman on horseshoe crabs, e.g. ‘Atlantic Horseshoe Crabs and Endotoxin Testing: Perspectives on Alternatives, Sustainable Methods, and the 3Rs’, Frontiers in Marine Science, 2020. I wonder if there could be a bit more reflection on some of the points made by Gorman (or in related work) - e.g. that some animals are out of sight in a regulatory and physical sense, and that makes their role in animal research less visible.

P13

-       ‘a consensus on the idea’ - ‘a consensus that’

-       ‘The same constraints…’ - I would incorporate this into the previous paragraph.

-       ‘scopes of this article’ - scope of

Author Response

Dear reviewer,
